# Advanced Bioactive Glasses: The Newest Achievements and Breakthroughs in the Area

**DOI:** 10.3390/nano13162287

**Published:** 2023-08-09

**Authors:** Maroua H. Kaou, Mónika Furkó, Katalin Balázsi, Csaba Balázsi

**Affiliations:** 1Centre for Energy Research, Institute of Technical Physics and Materials Science, Konkoly-Thege M. Str. 29-33, 1121 Budapest, Hungary; maroua.houria.kaou@ek-cer.hu (M.H.K.); furko.monika@ek-cer.hu (M.F.); balazsi.katalin@ek-cer.hu (K.B.); 2Doctoral School of Materials Science and Technologies, Óbuda University, Bécsi Str. 96/B, 1030 Budapest, Hungary

**Keywords:** bioactive glasses, glass preparations, bone scaffolds, implant materials, bioglass coatings

## Abstract

Bioactive glasses (BGs) are especially useful materials in soft and bone tissue engineering and even in dentistry. They can be the solution to many medical problems, and they have a huge role in the healing processes of bone fractures. Interestingly, they can also promote skin regeneration and wound healing. Bioactive glasses are able to attach to the bone tissues and form an apatite layer which further initiates the biomineralization process. The formed intermediate apatite layer makes a connection between the hard tissue and the bioactive glass material which results in faster healing without any complications or side effects. This review paper summarizes the most recent advancement in the preparation of diverse types of BGs, such as silicate-, borate- and phosphate-based bioactive glasses. We discuss their physical, chemical, and mechanical properties detailing how they affect their biological performances. In order to get a deeper insight into the state-of-the-art in this area, we also consider their medical applications, such as bone regeneration, wound care, and dental/bone implant coatings.

## 1. Introduction

Generally, three types of bioactive glasses can be distinguished, such as silicate-based (SiO_2_), phosphate-based (P_2_O_5_), and borate-based glasses (B_2_O_3_) [1,2]. Each type has different properties that can be exploited in many ways. The most attractive ways of using bioactive glasses as porous scaffold materials, drug delivery systems, and coatings on implants. The basic structure of bioactive glasses (BGs) is amorphous, they can bond to both hard and soft tissues, triggering new bone cell formation while degrading over time [3]. They possess outstanding biological activity resulting in stronger tissue or scaffold/implant interactions and bonds. In the form of porous scaffolds, they stimulate bone cell adherence and proliferation and their fast integration with natural bone tissues. In addition, their degradation rate is comparable with the rate of new bone formation kinetics and, additionally, prevents bacteria biofilm formation. To date, a large number of BGs have been developed based on silicate, phosphate, and borate that generally comprise mainly Na_2_O, SiO_2_, CaO, and P_2_O_5_ components in different ratios [3]. As a drug release system, these multifaceted materials are able to release different drugs that are previously incorporated into their porous structure. The amorphous glass structure with interconnected pore networks can absorb and release different bioactive ions, such as Ag, Ce, Co, Ga, Mg, Se, Sr, and Zn as well as therapeutic drugs [4]. On the other hand, dissolution products are particularly useful in wound healing applications also [3,5,6]. 

During the degradation process, dissolutions and precipitations take place at the interface in biological conditions. It is discussed that in silicate-based BGs, –Si–O–Si– bonds split up during the dissolution process by the act of hydroxyl anions, and silica ions are released into the biological solution. Meanwhile, CaP/HAp layer is deposited on the surface of the bioactive glass during precipitation, owing to the calcium and phosphate ions released from the BG materials and from the solution [7]. These processes strongly affected the biological performances of the bioactive glasses [8], since they can stimulate and improve cellular activity in the tissue healing phase because the released ions prompt biomineralization, thus aiding osteogenesis and angiogenesis, and even reportedly provided antimicrobial and anti-inflammatory effects [8,9,10]. 

Moreover, aside from the degradation process, the morphology, surface structure, and composition of BGs can also significantly affect cellular reactivity [10]. However, as is described in other works (based on in vivo experiments [11]) their degradation rate is still not fast enough for the perfect wound healing. These drawbacks gave rise to the development of other types of BGs, such as phosphate-based forms. These types of bioactive glasses can perform a faster solubility rate that can be adjusted and controlled by the composition of the glass [12,13]. Their dissolution mechanism is the hydrolysis of the P-O-P bonds, and the rate is highly dependent on the amount of the P_2_O_5_ component [14]. Phosphate-based BGs could completely degrade in a physiological environment. And the positive point is that the dissolved particles are already present in human bodies as essential or trace elements [15].

Meanwhile, the third class of bioglasses, the borate (B_2_O_3_)-based BGs have also emerged at the center of scientific attention [16,17,18]. Their main role so far is to repair specific bone defects, but they can also be used for the treatment of wounds. The preparation of borate-based BGs is done by changing the silica ions in the glass network to boron ions. Similar to the other types of bioglasses, at the surface of the borate-based bioglasses dissolution–precipitation mechanisms occur, and, as a result, cHAp layer forms. It is also reported that borate-based glasses can degrade faster than silicate-based ones and can transform almost entirely into cHAp. The dissolved ions will then act as growth factors in the cells [17]. According to several works of research [17], borate glasses are superior to traditional silica-based glasses owing to the poor solubility of the latter in body fluids. In addition, in the case of silica bioglasses, a spontaneous silica layer forms on the surface which causes incomplete transformation into an apatite. In addition, their higher tendency to crystallize narrows their biomedical applications [18]. Huang et al. [18] confirmed that the SiO_2_ substitution by B_2_O_3_ in commercial 45S5 bioglass resulted in faster hydroxyapatite layer formation on the surface of the bioactive glass. The degradation rate of boron-based bioactive glasses can be easily controlled and tailored by changing the composition and morphology in order to approach the rate of bone growth [19,20]. On the other hand, the sodium oxide content in the bioactive glass causes a higher inclination towards crystallization, limiting their shaping into different forms, and has some cytotoxic effect as well [21] which is caused by the dissolution of alkali ions into the physiological solutions or body fluids. According to the reports, the glass transition temperature (Tg) and the peak crystallization temperature demonstrated a linear decrease by increasing sodium oxide concentration. The sodium content also affected the thermal expansion coefficient as well as the density of bioglass. The performed preliminary in vitro biocompatibility tests confirmed that the glasses of higher sodium oxide content were responsible for the cytotoxic response. The measurement of the pH of solution revealed that the cytotoxicity was mainly owing to the ion exchange reactions at the glass surface. Bioactive glass partial crystallization also impedes its bioactive nature since the presence of crystalline phases limits the rate of the ion exchange reactions, namely the biodegradability in physiological conditions. The main crystalline phases found in the bioglasses are wollastonite, apatite, and phlogopite [21], of which the ratios and content were dependent on the original composition of the investigated bioglasses.

There are some attempts to solve this problem by removing the alkali oxide components from the bioactive glass structure [22], but this was difficult to implement because the complete removal of alkali oxides resulted in increased melting temperature and glass transition, reducing the vitrification range.

Another important innovation in bioactive glasses is their doping with different bioactive ions. These ions are essential for the proper functioning of the human body. The most common elements are Mg, Zn, Sr, Cu, and Mn [21]. Each of them has a different biological role and thus they can improve the bioactivity of the base bioglasses when applied in appropriate concentrations [22].

However, the proper doses of these bioactive elements or even drugs are still under extensive research and their clinical usage is uncommon. Thus, inspiring innovations in this field is essential to achieving a huge scientific breakthrough.

## 2. Main Preparation Methods for Different Bioactive Glass Structures

Two main different preparation methods are widely discussed for BGs, such as the melt-quench and the sol-gel. The applied post-treatment on samples is dependent on the demanded properties and their applications. It is noteworthy that, for biomedical usage, the purity of raw materials is crucial as well as the production of contaminant-free products.

### 2.1. Melt-Quench Method (MQ)

Oxide glasses are conventionally produced by melting the precursors (inorganic oxides, carbonates, fluorides, and others) in precious metal or ceramic crucibles at elevated temperatures, which are typically between 1200 and 1500 °C for bioactive glasses but it also depends on the composition itself. If the cooling is rapid enough, crystallization will be inhibited, and a glass structure will be obtained. In the traditional melt-quench technique they use oxide precursors and apply high melting temperatures with subsequent rapid cooling, so-called quenching. To acquire appropriate, homogeneous, and amorphous bioactive glasses, it is important to use proper crucibles, heating rate as well as dwell time during quenching. It is reported that melt-derived glasses can be post-processed by casting, chopping, or moulded into various shapes and morphologies [23,24,25]. The precursors in this method are the oxide form of the components, such as SiO_2_, CaO, and P_2_O_5_ which are mixed in different ratios according to the intended final products. 

### 2.2. Sol-Gel Method (SG)

The sol–gel method is a low-temperature and wet-chemical route to generating bioactive glasses. It can produce high specific surface area glasses with a large number of pores. The composition of the bioglasses can be easily adjusted to the specific requirements [26]. Usually, metal alkoxide precursors are being used in the process which transforms into an inorganic oxide network in either water or organic solvents [24,27,28]. 

Generally, the sol-gel method can be divided into three main steps, such as precursor solution preparation, the gelation process, and lastly the removal of the remaining solvents and salts by thermal treatment which can cause structural alteration [29]. In addition, these kinds of bioglasses can be more easily doped with bioactive elements or drugs to boost their biological performance than their melt-quenched counterparts [26]. The produced sol-gel can also be formed into different shapes and forms in various morphologies, such as monoliths, porous scaffolds, fibers, foams, and coatings [30].

Fiume et al. compared the physical properties of BGs prepared by both melt-quenching and sol-gel synthesis [31]. The same composition was used in all preparation techniques to aim for bone tissue regeneration. They reported that the bioglasses prepared by the sol-gel route had a specific surface area 2–4 times larger than the melted glasses. 

It is worth mentioning that recently more advanced methods have also come into sight to prepare different bioglass-based materials such as additive manufacturing, sputter coating, chemical vapor deposition (CVD), and 3D printing [32,33,34].

For silica-based bioactive glasses, mainly tetraethyl orthosilicates are used as precursors; however, many other types of precursors are also available [28]. The final form is a gel, which constitutes solid, condensate particles with a densely interconnected network of pores. Many parameters can affect the properties of the final bioglass, such as the precursor type, their ratio, the applied catalysts, pH, temperature, and even the environmental condition. In the sol-gel process, the gelling or polycondensation step is the longest, because this strengthens the network with stronger interconnected bonds. However, this process might cause shrinkage of the gel. Also, a very important step of the process is the elimination of the excess organic solvent from the final product.

Only a small portion of the scientific literature deals with the sol-gel preparation of phosphate- and borate-based bioactive glasses, which is ascribed to their more difficult chemistry and reaction mechanisms compared to the silicate-based ones. In the case of phosphate-based glasses, it was hard to find suitable precursors because the P-O-C bonds are strong and hard to hydrolyze, and thus the process is very slow [35,36]. Here, the P_2_O_5_ is the main glass-forming component, and the dominant structure of phosphate glasses is the orthophosphate ion. Phosphate glasses have diverse structural units. Interestingly, in a report, phosphoryl chloride (POCl_3_) was also proposed as a potential precursor [37] but the product was crystallized by the end of the process. 

In the case of borate-based glasses, borate alkoxides (B(OR)3) are typically used, they can hydrolyze quickly with boric acid precipitation, and then the boric acid hydrolyzes to form borate ions [38,39] 

Summarizing, the sol-gel method enables better physical and chemical variability of the final product because their properties can be easily tailored. Since the sol-gel-produced bioactive glasses have higher porosity and specific surface area, they can degrade faster and can accelerate hydroxyapatite formation on the bioglass surface [24,30]. As an innovation, Tuan et al. [40] prepared bioactive glasses by sol-gel by applying a hydrothermal system which effectively shortened the gelation time which normally lasts around a week. Their bioglass was a completely amorphous material with a mesoporous structure. Figure 1 demonstrates the steps of the two types of preparations in a detailed way.

## 3. Structural, Mechanical, and Chemical Properties of Different Bioactive Glasses

The mechanical and chemical properties and the structure of different bioactive glasses are crucial for their applicability. They must be highly porous with an interconnected pore-network, amorphous and, in a non-negligible manner, they also have to be mechanically durable, chemically stable, and biodegradable. In Figure 2, we show the three main types of the most commonly developed and used bioactive glasses highlighting both their required properties and possible application areas.

### 3.1. Silica-Based Bioactive Glasses

Porosity is the key factor when a bioglass is to be used as bioactive scaffold since it determines its mechanical performance and durability [41]. Baino et al. [41] aimed to study the elastic characteristics of silica-based scaffolds in terms of their porosity. They prepared highly porous bioglass foams via the sponge replica method and evaluated their elastic modulus, shear modulus, and Poisson’s ratio. In addition, they also determined the failure strength by compressive tests. They reported that when the total fractional porosity increased, the values of elastic and shear moduli decreased. They concluded that these innovative, highly porous silicate-based bioactive glasses could be potential candidates as bone grafts since their mechanical performances were commensurable to those of natural bones.

In the case of silicate-based bioactive glasses, various network modifiers are applied to stabilize the structure of the silica nanoparticles when they are produced via the sol-gel method. By changing the pH, temperature, and reagent concentration, the size of silica particles can be reduced to smaller than 100 nm; however, they can readily form larger aggregates [42].

For example, Aneb et al. [43] claimed that the ordered and controlled porosity of BGs, as well as the surface modification by silanization, can enhance the mechanical and biological performance of these materials. In the cases of BGs prepared by the sol-gel route, it is easy to use such surfactants as modifiers that can help to achieve patterned porosity and controlled pore sizes. These properties can noticeably improve their biological activity also. In addition, silanization, as a surface modifying method allows for the attachment drugs or other biomolecules, such as proteins to the surface. The paper concluded that by regulating the BGs’ porosity and applying surface modification, these newly developed functionalized BGs could be used as bioactive implants and drug carriers.

Today, thanks to the most recent developments, it is possible to produce BG materials as scaffolds that can provide sufficient support for bone regeneration in bone tissue engineering. The mechanical and chemical performance of these materials are highly dependent on the aging mechanisms as well [44].

According to Menci et al. [44], the aging step resulted in carbonate (Na_2_CO_3_ and CaCO_3_) and hydrocarbonate (NaHCO_3_) formations. The presence of carbonates improved the mechanical strength of the scaffold and reduced the pH, owing to the dissolved NaHCO_3_.

The scaffold should be designed to maintain controlled degradation kinetics in order to reduce the inflammatory response and to provide suitable mechanical properties during the healing process [45]. 

In the case of meld-quenched bioglasses, the sintering, performed above 1000 °C could consolidate the bioactive glass particles, thus providing the intended mechanical properties to the final product [46,47]. However, it is also reported that some commercial BGs (such as 45S5 and ICIE16) are difficult to produce into highly porous scaffolds with sufficient mechanical strength by sintering [48,49]. Due to the improper sintering, the BG powder would crystallize before it could densify sufficiently [50] and this might result in reduced bioactivity and poor mechanical strength [51]. In addition, since the amorphous part of the BG will preferably degrade, this results in scaffold structural instability. To overcome the problem caused by improper sintering, new types of network modifiers have been applied to reduce the crystallization rate. These modifiers are alkali oxides, such as K_2_O for Na_2_O [52,53,54]. On the other hand, a new type of silica-based bioactive glass in CaO-MgO-SiO_2_ structure with Na_2_O, P_2_O_5_ content, and CaF_2_ additives was developed by melting-quenching technique [55]. The effect of K_2_O and MgO addition on the base glass structure and on their thermal properties were discussed and the samples’ crystallization mechanism and mechanical properties were checked regarding their applicability, with these factors thoroughly reported in this paper. They found that the K_2_O addition resulted in higher Tg and crystallization temperature, but the MgO addition caused negligible effect. Moreover, the prepared glasses were prone to 3D crystallization and showed better machinability.

Reportedly [56,57,58], the substitution of disodium oxide with alkaline earth oxides (such as Ca, Mg, and Sr) in the traditional 45S5 Bioglass^®^ has been shown to be beneficial for improving the sinterability and mechanical properties of the glass maintaining optimal biocompatibility. In another research, Anghel et al. [59] assessed the crystallization stability of silica-based bioactive mesoporous bioglasses (BGs) with ceria addition in different concentrations. The scaffolds were prepared by the sol-gel route combined with the evaporation-induced self-assembly method (EISA). The XRD results confirmed that the main phases in samples were apatite, wollastonite, and ceria. The incorporation of ceria to the bioglass caused lower crystallization exotherm, and ceria segregation was observed. The ceria content decreased the crystallization tendency of the samples. According to the paper, these porous scaffold materials were meant to be suitable for hard tissue engineering.

Another study [60] discussed the effect of particle sizes of BGs on their overall reactivity. They claimed that the nano-sized BGs exhibited better bioactivity compared to the micro-sized ones, owing to their larger reactive specific surface area as well as faster ion release rate in biological conditions. The drawback was that the production of nanoparticles was difficult to achieve since usually the calcium was unevenly distributed into the silica network leading to lower Ca content, aggregations, and larger particle sizes. In this paper, the BG nanoparticles were produced by two techniques, such as reactive flash nanoprecipitation (RFNP) and the conventional sol–gel method. The results clearly showed that the sizes of the particles were smaller using the RFNP method with more homogeneous distribution and more even composition compared to the sol-gel route. On the other hand, in an earlier work [61], micro-sized SiO_2_-CaO-P_2_O_5_ ternary bioactive glass particle preparation was reported via the sol-gel route. They pressed the samples in tablet forms using dry press moulding. According to their analyses, the glass structure was amorphous, and the tablets were mechanically stable (with suitable mechanical strength) when soaked in biological solutions for two weeks. In addition, they also measured good apatite-forming ability of the prepared bioactive glass samples in vitro, immersed in SBF solution. The samples were proven to be biocompatible because the seeded osteoblast cells showed increased viability. 

### 3.2. Borate-Based Bioactive Glasses

The boron content of BGs has a profound effect on their mechanical and structural characteristics [62,63,64,65], and it also decreases their crystallization degree [66]. It was proven that the ideal boron content was around 2.7 mol%, and higher concentration caused deterioration in mechanical properties owing to the weak -B-O-Si- bonds [67]. Gharbi et al. [68] reported that with higher boron content the BGs became more thermally stable with a lower melting point. Lepry et al. [69] in their paper, published the effect of sol-gel preparation parameters of borate-substituted commercial BGs on their mechanical and chemical properties. They reported that a higher calcination temperature resulted in a partially crystallized glass and their specific surface area decreased noticeably compared to other types of glasses. The different precursors also influenced the gelation characteristics and the properties of the final product, but all the samples had high specific surface areas and porosities. In addition, the calcination temperature had the greatest impact on the mechanical properties since a glass–ceramic-like material formed. In addition, all the samples were highly bioactive, which was clarified by the rapid formation of cHAp on the surface of samples in biological solution. They also observed that the aging time and temperature had no effect on bioactivity. In conclusion, they confirmed that these types of borate glasses could be tailored as needed by altering the preparation parameters. The same research group [70] also investigated the effect of sodium content on the characteristics of borate-based glasses. They published that less sodium content inhibited the gel formation, and changed the morphology, however, did not cause crystallization. Moreover, the lower sodium concentration resulted in higher specific surface areas. Overall, the sodium content had a very slight effect on cHAp layer formation in SBF. Another research group [71], earlier, prepared lanthanum and strontium-containing borate-based bioactive glasses with the main composition of B_2_O_3_-SrO-Na_2_O-La_2_O_3_ and studied the effect of La on the glasses. They discovered that the La content stabilized the structure of the bioglass, resulting in better mechanical properties and they also observed a maintained Sr ion release from the BG when the Na^+^ was replaced by La^3+^. In a more recent paper, Aqdim et al. [72] prepared borate-based bioglasses by high-temperature melting and subsequent fast quenching method using Na_2_O, B_2_O_3_, CaO, MgO, and P_2_O_5_ powders as precursors. The characterization results confirmed that the boron addition in different concentrations and the gradual elimination of Na_2_O significantly affected the amorphous glassy structure. When the Na_2_O component was completely left out, the glass network formed interconnected large clusters of penta-borate and groups with four coordinated boron. The glass transition temperature shifted from 510 °C to 640 °C demonstrating a denser structure and reinforced glass network therefore better thermal stability.

The elastic modulus and compressive strength of some specific types of borate-based bioactive glass are exceptionally high [73]. A high concentration of boron is released at the implantation site from borosilicate glasses, which slows the degradation of biomaterials [73]. Additionally, they raise the pH of the growth medium. Polymer foam replication was utilized to create scaffolds out of glass ceramics [74,75]. The microstructure of these scaffolds was very identical to that of normal trabecular bone. Awad et al. [76] synthesized boron-based bioglass using the melt-quenching technique with a composition of 24.5Na_2_O-xSiO_2_-(45-x)B_2_O_3_-24.5CaO-6P_2_O_5_ to improve the glass bonding, speed up HAp deposition, and regulate dissolution. It was also discussed that the boron coordination number noticeably affected the structure and morphology of the BGs which are closely connected to their physical and chemical properties as well as their chemical stability. In some cases, [77] borate particles were incorporated into silicate-based BGs in order to decrease their chemical stableness. The preparation method reportedly had no effect on the final properties of the BGs. Deilmann et al. [78], on the other hand, claimed that the boron addition increased the hardness of BG, while no improvement was observed in the chemical stableness.

### 3.3. Phosphate-Based Bioactive Glasses

It is well-known that phosphate-based glasses contain ions that are already present in the human body, for example, calcium, sodium, phosphorous, and so on. These types of bioglasses are also bioactive and biodegradable. Their degradation rates can be controlled by changing their composition, morphology, and basic structure. The most ideal structure is when the bioglass is completely amorphous with many interconnected pores (either in nano, micro, or mezo size) [79]. For example, Jalilpour et al. [79] incorporated TiO_2_ into the phosphate-based bioglass and studied its effect on their stability and apatite-forming ability. The results revealed that the substitution of Na_2_O component with TiO_2_ accelerated spherical hydroxyapatite formation on the surface of the bioglass. In another interesting paper [80], the effect of CaO addition, into the novel phosphate-based glasses with a composition of Na_2_O-CaO-SrO-TiO_2_-P_2_O_5_, on the structure, thermal, durability, and bioactivity of the samples was thoroughly elaborated. The phosphate-based bioglass samples were prepared by the melt-quench method. They substituted CaO for the Na_2_O in the system which strengthened the glassy network. This meant that the glass transition temperature increased while the molar volume and the dissolution rate decreased. In another discussion [81], the phosphate-based glass with a composition of P_2_O_5_-CaO-Na_2_O-MgO exhibited an increase in glass transition temperature (Tg) with a decrease in crystallization degree when B_2_O_3_ was added at different mass percentages. The reason for the increase in Tg was that BO_4_ units were incorporated into the glass network and B^3+^ has a smaller cationic radius than Na^+^ [82]. Thus, the higher boron content resulted in a higher crystallization temperature due to chain length and cross-linkage density increase [83]. A very recent study [82] described the preparation and characterization of a phosphate-based bioactive glass with a unique composition of (60–x)NaPO_3_-25B_2_O_3_-xCaF_2_-15MgO, with x = 2.5, 5, 7.5, 10 and 12.5 mol%. Here, the samples were produced by melt quenching. The research group comprehensively examined the effect of intermediate oxides on the mechanical and structural properties, as well as the bioactivity of these glasses. They discovered that the CaF_2_ addition caused the sample to be denser and also initiated the crystallization of the fluorapatite layer. The role of the NaPO_3_ was a network former and it increased the mechanical durability of the glass. The findings also showed the relevance of calcium fluoride in adjusting the chemical stability and bioactivity of these glasses. Recently, Li et al. [83] developed a novel mesoporous phosphate bioactive glass with a structure of P_2_O_5_-CaO and P_2_O_5_-CaO-Li_2_O by the sol-gel method using surfactant Pluronic P123 as a pore-forming component. The results showed that this surfactant was a good inhibitor of crystallization. They also revealed that the increase in CaO content in the phosphate-based BG structure had no noticeable effect on the phosphate chain structure and mesoporous nature, but the specific surface area and pore volume slightly increased. Similarly, the Li_2_O addition did not cause any change in the bioglass structure, morphology, and porosity characteristics. The research group observed that the Li content above 20 mol% generated crystallizations. The performed bioactivity tests showed relatively low acellular activity for all samples with slow kinetic HAp deposition onto the surface of BGs. In this case, the main role of lithium was inducing osteogenic differentiation [84]. Moreover, it is widely discussed that Li can protect cartilage from degradation caused by cytokine [85]. Therefore, Li-containing BGs are suitable for Li-ion delivery systems in bone and cartilage tissue engineering. We outline the major benefits and drawbacks of the different types of BGs in Table 1.

## 4. Biological Responses of Different Bioactive Glasses

### 4.1. Silica-Based Bioactive Glasses

The silica glass composition precisely determines the biological properties of the material as well as the way how they will act in vitro and in vivo [8,86]. 

As its main component, the silicon is present in Si^4+^ form in biofluids and is reported to be essential for the metabolic activity of bone cells during new bone growth. According to previously presented studies, silicon is a key component of glycosaminoglycan and its protein complexes, which are abundant in bone and connective tissue and may have an impact on bone growth and maintenance [87]. In vitro studies have also revealed that silicate ions boosted the osteoblast cell proliferation and differentiation while also showing angiogenic and osteogenic effects [88,89,90]. Calcium is another crucial constituent of the bioglass which plays an important role in the extracellular matrix (ECM) mineralization and bone cell growth. The Ca content in the bioglass scaffolds is beneficial for the adhesion, proliferation, and differentiation of osteoblast cells thus helping in bone regeneration [89]. Additionally, in a non-negligible manner, the phosphorous part of the bioglass is important in many biological processes, such as bone growth and ossification or re-building [8,84]. The degradation mechanism of bioactive glass-based scaffolds in SBF has already been previously described by Boccaccini et al. [50]. Based on the described model, scaffold degradation happens at the interface between the bioglass and the Na_2_Ca_2_Si_3_O_9_ crystallites (which is a rare form of silicate mineral) and results in the degradation of non-sintered crystalline structures. The ion exchange (Na^+^ and Ca^2+^) during the degradation caused amorphization and a rapid increase in pH. However, owing to the buffering ability of the applied SBF solution, the increase in pH values could be slowed down. This buffering feature is important in the case of in vivo applications because the high pH causes cell death. Menci et al. [44] evaluated the bioactivity of glass-based scaffolds by immersion them in SBF. The weight reduction measured in the cases of the prepared scaffolds after immersion was due to their partial degradation and the dissolution of carbonate particles. With increased aging time, they observed greater weight loss. 

In vitro and in vivo studies are fundamental steps in the characterization of new implantable materials to preliminarily assess their biological response. To prove this fact, Fiume et al. [91]. presented the in vitro and in vivo characterization results of a novel silicate bioactive glass with the following composition: 47.5SiO_2_-10Na_2_O-10K_2_O-10MgO-20CaO-2.5P_2_O_5_ in mol.%. The cytocompatibility of samples was assessed with human mature osteoblasts (U2OS), human mesenchymal stem cells (hMSCs), and human endothelial cells (EA.hy926). They observed perfect pro-osteogenic characteristics of this type of bioglass. They obtained a statistically significant difference in bone formation in vivo in comparison with the control (untreated) group and the experimental one, revealing a good osteogenic impact caused by the implanted bioglass at the defect site. The implanted bioglass reportedly dissolved completely after only 3 months and was replaced by newly formed tissues. These results have clearly proven the great osteostimulatory capability of this bioglass. Earlier, the same group [92] reported the results of the in vitro and in vivo biological response of a silica-based bioactive glass with the following composition: SiO_2_-Na_2_O-K_2_O-MgO-CaO-P_2_O_5_. The in vitro experiments showed excellent adhesion of the investigated cell lines onto the surface of BGs, and they had better cytocompatibility compared to the control group. On the other hand, the in vivo tests revealed a statistically significant difference between the rate of new bone formation between the control and the experimental groups at all investigated implantation periods. The prepared BG demonstrated an osteogenic effect and resorbed totally within 3 months following implantation. 

However, in other works [21,93], it was demonstrated that the bioactive mechanism of degradation depended on the ionic dissolution of glass which led to osteointegration and faster regeneration.

During the development of new bioactive glasses, in vitro cell culture studies serve as the first test of the response of a biological system in direct contact with the material. These tests could give valuable and reliable information about bioactivity, cytotoxicity, and even genotoxicity through the measurements of cell proliferation and differentiation [94]. The parameters measured in the cell culture experiments have to be properly selected according to the intended use of the selected BG composition, for example, the cell types and the specific assays to be utilized [95]. There are also very recent works evaluating the incorporation of different therapeutic agents (such as Sr and Cu) into silicate-based bioactive glass with the base structure of SiO_2_–CaO–Na_2_O–P_2_O_5_ [96,97]. Some researchers have also added SrO and CuO into the silicate bioglass in different concentrations and ratios and evaluated their effect. Here, the key role of the Cu element is to help the function of important enzymes in bone formation and re-building [98,99]. The bioactive glass was incorporated with strontium, which is also an essential trace element in the body and has a remarkably similar chemistry to calcium [100,101]. As a conclusion, the results revealed that Sr-containing bioglass had a higher crystallinity and larger particle size of apatite after immersion in SBF, on the other hand, the Sr and Cu co-incorporated bioglass showed lower crystallinity and particle size. They attributed this phenomenon to the competition between Cu^2+^ and Ca^2+^ ions during apatite precipitation. In addition, the Sr content also accelerated the degradation process. On the contrary, the Cu content lightly reduced the crystallization rate after SBF immersion due to a slower dissolution of the glass. Nevertheless, both types of bioglasses were appropriately bioactive.

Very recently, another research group [102] has focused on the preparation and biological evaluation of Cu, Co, and Sr-containing bioactive glasses. They prepared the bioglasses via the sol-gel route using SiO_2_, CaO, P_2_O_5,_ and SrO precursors. They followed the in vitro bioactivity response of the glasses for 14 days in biological conditions. The results indicated good cytocompatibility as well as good osteogenic ability of the investigated samples. After immersion in SBF, they also observed HAp and polymorph calcium carbonate deposition on the surface of BGs. Interestingly, the cobalt-containing BG sample demonstrated better biocompatibility, and faster bone nodule formation was perceived compared to the other BG samples. 

Some previously reported studies [103,104] have also proved that Sr-doped 45S5 glasses had stronger bone-attaching capability than the bioglasses without strontium. Copper is another widely used dopant and essential trace element that can enhance even neovascularization and, in addition, has a slight antibacterial effect [105]. The other previously reported advantageous properties of Cu^2+^ ions are that they can enhance the proliferation rate of many cell lines, such as bone cells, osteoblasts, endothelial cells, fibroblasts, and so on, in vitro [106], and could increase the vascular endothelial growth factor expression in vivo. As for the cobalt addition, it is proven that it can promote the angiogenesis process and is also a favorable constituent in bone tissue re-modeling and bone repair [107,108,109]. The copper element also has a positive effect on angiogenesis and bone regeneration both in vitro and in vivo [110,111]. Atkinson et al. [111] developed novel, mezoporous zinc oxide containing bioglass systems with composition of 70SiO_2_–(26-x)CaO–4P_2_O_5_–xZnO (x = 0, 3 and 5 mol%) by sol-gel and polymer-template method and evaluated their structural, biological, and antibacterial properties. The samples turned out to be bioactive with excellent apatite forming ability, and crystalline calcite phase was also formed at the surface of the samples after two and four weeks of immersion in SBF. In addition, they discovered that the higher zinc content in the bioglasses decreased the tendency of calcite formation because of the decrease in calcium ion concentration in the SBF solution which hindered the HAp generation in the beginning but had no long-term effect on the HAp deposition afterwards. The zinc content provided excellent antibacterial effect and inhibited the growth of B. subtilis and P. aeruginosa strains.

Another research group [112] investigated the osteogenic characteristics of the dissolved bioactive ions from mesoporous bioglasses (with base composition in mol%: 70SiO_2_, 30CaO) with and without Zn and Cu doping. To assess their biological activity, human bone marrow-derived mesenchymal stromal cells were used. They found that the Zn-containing bioglass had a positive effect on the cell viability compared to the undoped ones, while they measured a slightly increasing tendency in the case of Cu-containing BGs at the later stage of the osteogenic differentiation as well as on calcification and even extracellular matrix formation.

In a novel approach [113], tellurium dioxide was used to provide antioxidative and antibacterial features to the base bioactive glass. The characterization results confirmed that the tellurium addition did not cause a change in the amorphous glass structure, while the increased amount of TeO_2_ compared to SiO_2_ resulted in Tg decrease. The in vitro bioactivity test clearly showed that the Te-containing glass could readily induce HAp deposition on the surface of the glass substrate. In addition, the sample also had more efficient antibacterial and antioxidant effects compared to bioglass without tellurium. 

### 4.2. Borate-Based Bioactive Glasses

The borate-based BGs are less common in use than the traditional silica-based glasses [25], even though they have shown outstanding bioactivity in in vitro studies using various compositions [114]. It has already been demonstrated that borate-based glasses can undergo a more rapid conversion to carbonated hydroxyapatite (cHAp) in vitro [19] and in vivo [115,116,117,118] compared to silicate-based glasses, which is attributable to their higher solubility. Lepry et al. [119] studied the effect of in vitro dissolution media on the bioactivity of sol-gel-derived bioactive borate glasses. They used six different media, such as SBF, tris(hydroxymethyl)aminomethane (TRIS, pH 7.4), Dulbecco’s phosphate-buffered saline (PBS), Dulbecco’s Modified Eagle Medium (DMEM), saline, and deionized water in their work and they observed a rapid increase in pH due to the glass dissolution and ion release. They also reported that the best apatite-forming media would be the SBF, TRIS, and PBS, and the media that generated calcite layer deposition were the DMEM, saline, and deionized water. In more current work, the same research group [120] studied the effect of magnesium addition on sol-gel-derived bioactive glass compositions. It is well known that the addition of magnesium to the structure contributes to numerous important tissue functions and can promote their repair. The effect of ionic release was evaluated using MC3T3-E1 pre-osteoblastic cells. They found that increasing the MgO content had no significant effect on the glass structure; however, the textural characteristics drastically changed in the case of samples with more than 20 mol% MgO content, since they showed reduced specific surface area and pore volume values. The higher MgO content also worsened the bioactivity of the bioglass, and the ionic dissolution products were cytotoxic to the MC3T3-E1 cells. In the current work of Ishihara et al. [121], borate-doped silicate glasses were prepared by sol–gel technique as a new candidate in tissue engineering. They investigated the effect of borate concentration within the glass structure on the morphology, ion-dissolution rate, and fibroblast cell viability. They reported that the higher boron content caused a decrease in the specific surface area of the glasses and the calcination temperature increased from 600 °C to 700 °C. The decreased surface area also affected the ion release properties of these bioglasses. In addition, as a positive biological response, the proliferation rate of the investigated fibroblast cells cultured with the dissolution products from the glasses increased significantly with increased borate content. The cytocompatibility was assessed by measuring the metabolic activity of the mouse fibroblast-like cell line. The results demonstrated that the calcium borosilicate sol–gel glasses could be more effective for the enhancement of cell activities and following tissue regeneration in the body compared with calcium silicate glasses. In conclusion, it was possible to control the ion release rate of the base calcium silicate bioglasses through appropriate borate addition. Meanwhile, the slower ion release achieved suggested that these types of bioglasses are also suitable to prevent any inflammatory side effects maintaining their beneficial biological response. Theoretically, during tissue regeneration and in order to accelerate the healing, a governed and sustained ion release from the BGs is necessary since the majority of cell responses depend on the number of ions moving toward the cells. Some researchers claimed that during the early phase of immersion, the ion release rate is faster than it would be desirable, and this could cause negative effects on cells. Moreover, the increased pH, therefore, can cause an inflammatory response also [122]. In a remarkably interesting and recent paper, Alasvand et al. [123] developed a new type of bioglass that can be used as a blood-contact biomaterial. To date, this kind of application was confined by the fact that the BGs have a negative effect on the activation of the coagulation pathway. Modified borate-based BGs with a composition of B_2_O_3_, MgO, K_2_O, Na_2_O, and different amounts of CuO were produced by a melting-derived route. These BGs provided an anticoagulant effect because they could eliminate the blood-clotting elements, such as calcium, phosphate, and silica. According to the report, these BGs were in the amorphous phase and had no cytotoxic effect. On the contrary, the modified BGs exhibited an antibacterial effect against bacteria. Hemocompatibility tests were also performed on samples and the results proved that none of the samples caused red blood cell death nor triggered the coagulation pathway. Moreover, the addition of CuO into the BG structure significantly enhanced the endothelial cell proliferation, migration, vascular endothelial growth factor secretion, and so on, proving improvement in the angiogenetic properties. They concluded that these types of multifunctional BGs are suitable for blood-contact applications as well as for vascular tissue engineering.

### 4.3. Phosphate-Based Glasses

Phosphate-based glasses are exceptional in a way that they can be exploited in both hard and soft tissue engineering. Moreover, they are very prospective candidates for many biomedical applications owing to their inherently porous microstructure and suitable morphology [25,79].

Li et al. [124] examined how the phosphate content induced faster osteogenesis both in vitro and in vivo. They have discovered that higher phosphate concentration in the bioglass system significantly increased the apatite-formation ability. They prepared four types of phosphate-based bioglasses through the melt-quench route and studied their apatite-forming capacity in α-MEM culture medium. The BGs showed enhanced cell viability values and alkaline phosphatase activity of osteoblasts by increasing the phosphate content. The in vivo study performed confirmed a larger amount of new bone generation in the calvarial defects using high phosphate-containing BG granules as implants rather than that of BGs with lower phosphate content after 8 weeks of the surgery. As a further development, Babu et al. [125] prepared Zn-incorporated bioactive phosphate glasses with the aim of enhancing their bioactivity and antibacterial characteristics. These glasses could promote bone in-growth and thus, they were claimed to be efficient as resorbable implants in bone repair. The bioactivity of samples was tested in simulated body fluid. The results revealed HAp layer formation on the phosphate-based bioglass which is a good indicator of bioactive characteristics. They also followed the degradation of the samples and discovered that the ZnO concentration within the bioglass affected the degradation rate and characteristics. The performed in vitro cytocompatibility tests which showed increased cell viability and proliferation. They did not observe any cytotoxicity of the ZnO, however, it had an antibacterial effect against the investigated bacteria.

## 5. Application Possibilities of Bioactive Glasses

The main aim of the clinical application of different bioglasses is to help successfully repair damaged bones as well as to accelerate the healing processes [126]. The most commonly used materials in clinics so far are autografts, allografts, xenografts, and bone cement [127]. However, conducting any clinical applications is overly complicated due to the relatively high possibility of negative immunogenic responses and body rejection [128]. 

### 5.1. Bioactive Glasses in Soft and Hard Tissue Engineering

Numerous research works have been conducted so far to develop bioactive artificial bone replacements that can be applied directly to the injured part and stimulate osteogenesis on-site [129]. BGs are suitable candidates for the treatment of hard tissue/bone damage owing to their above-discussed outstanding, diverse, and adjustable properties [129,130,131,132,133,134]. It is also important in tissue engineering that the BGs enable to bond properly to both hard and soft living tissues [132]. The silicate-based bioactive glasses are typically used in a granulated form in bone regeneration. As scaffold materials, they are difficult to prepare, because of the higher rate of crystallization during sintering. However, the right adjustment of sintering parameters allows for porous amorphous scaffold preparation [133] with sufficient mechanical strength in bone tissue engineering. In the work of Aalto-Setälä et al. [133], amorphous scaffolds with approximately 50% porosity were prepared by optimizing the sintering parameters. This scaffold had compressive strength close to that of cancellous bone and so, it could be used in surgical applications. This type of scaffold was able to support 3D bone ingrowth in rabbit femurs [134] which further confirms their applicability as scaffolds in load-bearing bones. In current work, Chen et al. [135] reported fluoride-added alkali-free bioactive silicate glasses as bone substitutes to repair any bone defects or treat bone losses caused by different diseases. The fluoride content in their case accelerated the osteogenesis in vitro, but the halide content increased the rate of degradation and the apatite-forming ability. It is noteworthy that these halide-containing BGs caused faster osteogenic differentiation of investigated cell lines and induced faster bone nodule growth and collagen generation in vitro, while the in vivo tests proved that the halide-containing BGs speeded up the bone regeneration process. They also discovered that fluoride and chloride had synergistic effects on osteogenesis and angiogenesis both in vitro and in vivo. Taking all the results into account, these types of BGs are very suitable for bone substitutes. In addition, by changing the concentration of fluoride and chloride in the glass structure, the properties of BGs can be adjusted to the specific requirements in clinical use. It is widely researched that when the BGs are applied as porous 3D scaffolds/bone grafts, the chosen preparation method is also relevant because it precisely determines the morphology, porosity, and such mechanical, chemical, and even biological performance of the material, affecting the cell migration, adhesion, bone cell ingrowth, new bone formation and so on [136]. To conduct an in vivo experiment, many factors have to be considered, such as the chosen animals, the defect type, size, and location in the body, as well as the period of implantation. The main risk of implantation of BGs is the possible potential toxicity of their degradation products, which can easily cause many side effects [136,137,138]. In a study, Gestel et al. [139] reported on the use of BGs as granules to cure osteomyelitis. The BGs investigated by the research group have shown good clinical results. They cured the infected bones with large defects by applying these BGs as load-bearing bone graft materials. They also reported that these BG structures could ease the load on the cortical bone which was weakened by a large defect. 

There are elaborate works on borate-based BGs, which demonstrate that these materials can be employed for in vivo bone repair in the same way, without any cytotoxicity or side effects [38,73,140,141], since a low amount of boron can promote osteogenesis, while the controlled release of boron ions during degradation hasten the bone repair mechanism [73]. In addition, it has been discovered that bioactive borate glasses have higher degradation rates, making them more suitable candidates in both hard tissue and soft tissue regeneration than silicate-based ones. However, the constitutions of these borate glasses are still dependent on the traditional silicate glasses, and their chemistry is still not optimized and monitored. While in the cases of silica-based glasses the silicon tetrahedron gives the primary structural unit, their borate-based counterparts are built up by planar, trigonally coordinated BO_3_ groups to form vitreous borate glasses [142]. As a beneficial characteristic in tissue engineering, the borate-based BGs can control the ion release processes (such as of Ca^2+^ and other bioactive or therapeutic ions), and thus can provoke osteogenic gene expression. Similarly, these bioactive glasses direct the delivery of drugs and nutrients which is essential for new bone formation [38,143]. It is also reported that scaffolds prepared from borate-based BGs exhibited better bioactivity compared to silica-based BGs, which was attributed to their controlled and gradual decomposition in biological environments [144]. Experiments have proven that the angiogenic and osteogenic gene expressions of different cell lines were increased by the boron-based BG scaffolds [145,146]. Some researchers clearly claim that the boron-based BGs can be superior to silica-based ones and a better choice for the treatment of damaged bones since they can be fully conversed to cHAp; however, to date, there is no such product available on the market [147]. To improve their biological performance even more and make them able to act against bacteria, these bioactive glasses were doped with the following metal oxides: zinc oxide, tellurium oxide, silver oxide, titanium oxide, and gallium oxides. Mutlu et al. [148] reported that ZnO is more effective against *Staphylococcus aureus* than Ga_2_O_3_. Additionally, the borate ions can be readily utilized to aid in wound healing since they reportedly advance cell proliferation and differentiation, promote angiogenesis, and even encourage extracellular matrix formation [149].

Other very prospective and interesting utilization of BGs is as injectable bone cement. These materials have numerous advantages, such as easy and fast preparation, low cost, suitable form and viscosity for many clinical applications, appropriate mechanical characteristics such as micro- and macro-porosity, excellent bioactivity, good integration ability in the body, and no side effects. Last, but not least, bioactive ions and therapeutic agents can be easily added into their system. The perfect mechanical and rheological properties are achieved when the BG particles are mixed in different biopolymers with different ratios [150,151]. 

The BGs can be incorporated into natural and synthetic polymers as binders in the same way to obtain an injectable suspension/paste with various viscosity [152,153]. Commonly used polymers can be for example chitosan (an amino-polysaccharide), [154,155], collagen [156,157], gelatine [158], sodium alginate [159,160], cellulose derivatives [161,162,163], poly (lactic-co-glycolic acid) (PLGA) [164] polyethylene glycol [165], and PMMA [166]. The applied polymers caused slightly reduced porosity, but this fact did not cause a considerable effect on their biological performances either in vitro or in vivo [167]. One perfect example was demonstrated by Zhang et al. [156]. They described the preparation of a new generation injectable bone graft/cement that was bioactive and increased the osteogenic activity, resulting in faster osseointegration. This novel bone cement was strontium-doped borate bioactive glass particles incorporated in a chitosan-based bonding matrix. In this case, the role of the BG particles was to promote osteogenesis, while the chitosan matrix served as a degradable adhesive. In an in vivo study, they observed new bone formation in the vicinity of the developed BG implants in two months in a critical-sized rabbit femoral condyle defect model. The overall results indicated that this type of bone cement is perfectly suitable as injectable grafts for complex-shaped bone defects applying minimal invasive surgery. To prove the importance and actuality of these types of BGs, a very recent paper [155] also presented the applicability of injectable strontium-doped mesoporous bioglasses as a treatment for osteoporosis. The developed composite material was prepared by the addition of the strontium-bioglass particles into calcium sulfate powder and then mixed into PLGA polymer matrix. These samples showed anti-osteoclastogenic features and the measured mechanical properties and cement setting time were appropriate for clinical application. They concluded that these materials had the ability to induce bone regeneration and inhibit bone resorption. Figure 3 illustrates the commonly used allopolast in bone tissue engneering.

Bioactive glasses are used in a specific way in soft tissue engineering, as is described thoroughly in a recent review paper by Mazzoni et al. [168]. The composition and structure of BG scaffolds for this purpose can be tailored to sufficiently simulate the structure of the aimed tissues. Thus, they can be used to accelerate the regeneration of damaged tissues by aiding new tissue formation [169]. There are undergoing and promising investigations of BG-incorporated biopolymers for skin, muscle, and gum regeneration [170], as well as for evidence of their efficiency in cardiac, lung, epithelial, and nervous tissue repair [171] as hydrogel composites or electrospun fibers. According to the overview of Kargozar et al. in 2020 [172], they described that all three types of BGs are effective both in vitro and in vivo in nervous system healing. In addition, the mixing of BGs into conductive biopolymers enhanced the neural cell performances at the damaged sites. To prove this, Lin et al. [173]. recently developed collagen-polypyrrole copolymers containing Sr-added nanoBGs by electrospinning and assessed their in vivo performance in nerve cell regeneration. As a result, these composites were suitable bases for nerve cell growth and proliferation and provided faster wound recovery. In other work [174], poly(glycolic acid) (PGA)/collagen/nanoBG electrospun fibers were produced as materials for nerve conduits. The bioglass content improved the biological, mechanical, and chemical performances of nanofibrous conduits. On the other hand, earlier, Souza et al. [175] prepared double-layered conduits with BG microfiber-incorporated polycaprolactone (PCL) fibers, as an innovation. They proved that the developed samples were water-permeable membranes, hence promoting the metabolite exchange of nerve cells and the adjacent environment. Epithelial tissue recovery is similarly a crucial issue in soft tissue engineering since these tissues are the main constituents of the body skin and cover the external surfaces of all organs and cavities. Thanks to the huge effort devoted to the development of new types of BG composites, these can be very useful during the wound healing process, especially when they are incorporated with antibacterial agents to prevent infections [176]. Hence, they are appropriate for wound dressing or as specific ointments (proven by clinical studies) [176,177]. The most extensively examined and assessed BGs are the silicate-based glasses so far owing to their increasing effect on pH on-site, and their proven antibacterial effect. These specific properties are attributed to the Si ion release that is accountable for collagen stimulation and neovascularization. Similarly, borate-based glasses exhibit great expectations in wound healing implementation due to their high dissolution degree causing a rise in local pH, and the ability of boron to stimulate angiogenesis [178]. The biodegradable boron-based BGs reportedly induced rapid wound closure in animals and also in diabetic patients [178], and the phosphate-based BG structures reportedly have huge potential in facilitating neuron regeneration after nerve injury besides their outstanding wound healing ability [179]. In addition to wound healing, BGs are also a point of interest for the repair of other epithelia, such as the respiratory and gastrointestinal epithelium [4,177] and for cardiac and pulmonary tissues, as is extensively described in the review by Kargozar et al. [180]. In this case, the main reactions of the BGs are the dissolution and precipitation at the implants’ surface that are in contact with the body fluids, such as human plasma. These reactions can advance the formation of a strong bond between the bioglasses and the given tissues, and the collagen content in the soft tissues can ease the adherence to the surface of BGs [181].

The most ideal form of bioglasses in these cases are injectable hydrogels, in which BGs and repair cells are embedded in biopolymers, and patches made by electrospinning, bioprinting, or molding. These patches are also loaded with GS and the required cell types are being increasingly investigated and developed as effective tools to regenerate the damaged tissue [182,183,184]. In Figure 4, the possible applications of BGs and BG composites are demonstrated.

### 5.2. Bioactive Glasses as Drug Carriers

BGs are recognized as versatile materials in biomedicine and can be exploited as drug carriers as well, owing to their highly porous nature which allows for encapsulation and controlled release of bioactive elements or any therapeutic agent. One of the most interesting application possibilities lies in cancer therapy [185,186]. In addition, there is also huge potential in the development of different antibiotic or drug loaded mezoporous bioglass (MBG) scaffolds for biomedical applications. For example, a recent study [187] proposed an innovative material to improve the quality of lives of people suffering from traumas, bone diseases, or even aging. The research group developed cerium and vancomycin-loaded MBG on polymethyl methacrylate (PMMA) sacrificial template. The role of the cerium content was to provide sufficient antibacterial effect and tumor therapy while the vancomycin hydrochloride acted as a commonly used antibiotic. The antibacterial activity and drug delivery efficacy of samples were evaluated, and the results revealed positive effects on the treatment of damaged bones. Obviously, the vancomycin containing samples showed higher antibacterial activity, and the in vitro cytocompatibility test did not show cytotoxicity on the investigated L929 cell-line. The drug release profiles had two discernible stages: a fast release stage, followed by a slower sustained release rate. On the other hand, the cerium content caused some decrease in the vancomycin release rates, which can offer a prolonged dissolution process. In other research work [145], bisphosphonate/zoledronate has been loaded onto the BBG carrier, and it has demonstrated good mineralization in both in vivo and in vitro tests. The authors compared the efficacy of biodegradable boron-based bioglasses with the calcium sulphate that had already been used in clinics for the treatment of osteomyelitis and as a carrier substrate for vancomycin drug. The results proved a much better performance of the vancomycin-loaded BG samples compared to the other groups against the induced bone infection. In a newly published manuscript, [188], Ta_2_O_5_ and ibuprofen-loaded boro-phosphate glasses were developed and studied regarding their bioactivity and drug-release capacity. The results clearly proved that the drug-containing boro-phosphate glass had no toxic effect on bone cells; they were sufficiently bioactive and demonstrated a sustained long-term drug release. Another group [189] recently published a paper about alendronic acid (ALN) and flufenamic acid (FA) loaded mesoporous bioactive glasses and calcium phosphate cement and reported their biological performance, such as drug release rate, bioactivity, and biodegradability. They observed a sufficiently slow drug leaching rate, and the developed drug-carrier composite fulfilled the standard requirements for clinical usage they claimed these drug-loaded composites possessed as a prospective candidate in osteoporotic bone-filling surgeries.

In a way, the strontium element can also be regarded as a drug, since it has already been proved in many in vivo studies that it can enhance the mechanical strength of bones [190], and act as a potential therapeutic agent for osteoporosis treatment [191]. For this reason, MacDonald et al. [192] examined a new type of bioglass (borate glass-reinforced resin composite with hydrophilic modification) doped with strontium. The composition of the base borate glass in this case was as follows: 70B_2_O_3_-26SrO-4Na_2_O (in mol%), and it was mixed with different resins to obtain composite materials. They studied the strontium ion release profile over a long time. They reported that the Sr and B ion release ratio was dependent on the applied resin which was attributed to the strontium phosphate precipitation onto the surface of the composites. The composites produced a continuous and sustained Sr release at the level sufficient for osteoblast proliferation. It is also well known that Sr-loaded bioactive glasses can stimulate bone cell in-growth by inducing osteoblasts cells and inhibiting the osteoclast cells from bone resorption and, as such, they can be a potential treatment for osteoporosis. The other essential element is silver, which is knowingly the best and only broad-spectra antibiotic without bacteria resistance. Silver ions have a similar size to sodium ions, thus they are interchangeable within the bioglass structure [193]. They also present an identical dissolution behavior/rate that causes the antibacterial effect. This effect is important in clinical use because it is the main cause of implant failure or tissue necrosis after infections [194,195]. Many studies [196,197] have stated that silver-added bioglass materials had bactericidal/bacteriostatic effects and, in the meantime, they maintained the bioactive properties also. Other antibiotics are also under investigation for incorporation into different bioglass systems, such as third-generation antibiotics, including ceftriaxone and sulbactam sodium (CFS) [198,199], and even tobramycin [200].

The thorough research and examination results clearly revealed that the incorporation of bioactive ions, biominerals, and other therapeutic agents or drugs has a profound effect on the biological performances of all types of bioglasses either in the form of porous scaffolds, coatings, or composites. We list the doping agents and their effects that have been reported so far in Table 2.

### 5.3. Bioactive Glasses in Dentistry

The main aim of bioglasses in dentistry is to fill in the voids in the damaged bones and teeth (bone fillers or bone grafts) or in osteoplasty. The traditional silica-based bioglasses can degrade gradually whilst releasing different ions, such as Si^4+^, Ca^2+^, PO_4_^3−^, and Na^+^). During degradation, they also form an apatite layer when in contact with body fluids. The traditional, first developed, and commercialized bioglass (45S5 Bioglass^®^) has been widely used in orthopedics and dentistry since 1985 until date now [132], but its apatite-forming ability is relatively slow. However, increasing the phosphate content in BGs caused faster apatite deposition and better integration [124,201] compared to the original one. Some oral diseases (such as periodontitis, peri-implantitis maxillofacial defects, and so on) often cause bone loss or defects. These diseases greatly deteriorate the patient’s quality of life. Bone grafts (autologous and allogenic) are commonly used treatment methods. But they have some major drawbacks, such as contaminated source/raw materials and rejection by the body’s immune system [124,202,203,204]. There is a huge need for alternative artificial bioglasses that have perfect bioactivity and can easily integrate into the body, accelerating osteogenesis [205]. A research group [206] prepared pure and silanized, calcium-containing conventional 45S5 bioglass and Sr-containing 45S5 bioglass as fillers for dentins. They evaluated the samples’ mechanical and bonding strength along with the remineralization and collagen degradation rate. Based on the results, the incorporation of silanized bioglass into commercial dental adhesives provided a beneficial effect regarding the dentin remineralization process, meanwhile strengthening the bonding by aiding the adhesive polymerization. However, the strontium content weakened the microtensile bond strength of samples, so the changing calcium to strontium in the bioglass structure negatively affected the capability of commercial adhesives to create a sufficient strength dentin bond. Another research investigation, [207], was dedicated to investigating the effect of bioactive glass particle addition on the acrylic resins commonly used in dentistry. Acrylic resins are usually used in prosthetic dentistry owing to their long-term durability and easy preparation technics [208], as well as their acceptable biocompatibility [209]. However, their bioactivity could be improved even more by adding bioactive particles into their structure, such as different bioglasses in different ratios. Unfortunately, besides several advantages, acrylic resins possess some disadvantages, such as the restricted saliva flow within the denture base area. Raszewski et al. [207] assessed the samples’ mechanical strength, sorption ability, and solubility based on the ISO 20795-1:2013 protocol. The bioglass-containing samples demonstrated lower compressive strength than that of pure polymethyl methacrylate but still met the standard requirements. 

In current research work [210], porous 3D scaffolds were developed with an innovative composition based on phosphate-based bioactive glass with the addition of β-tricalcium phosphate and zirconium oxide at different concentrations and ratios. They produced the scaffold using a combination of powder metallurgy and polymer foaming. Such prepared scaffolds were mechanically stable with sufficient compressive strength and elasticity modulus very similar to human trabecular bone. They tested the biological performance of the scaffolds in vitro by immersing them in artificial saliva and they observed CaP precipitation on the surface with a flower-like morphology that did not contain zirconia. However, the zirconia-containing samples demonstrated slower apatite forming ability and faster solubility.

There are also attempts to utilize bioactive glasses as dental prostheses and orthopedic implants [211]. Usually, in this case, the bioactive glasses were prepared by melt-quench route with and without TiO_2_ addition. According to the findings, the TiO_2_ addition to bioactive glass compositions noticeably affected their optical and radiation characteristics. Another specific usage of BGs is as air abrasives. An interesting paper [212] described and discussed the fluorapatite formation ability, degradation rate, and the biological safety, as well as cutting efficiency of the fluoride- and chloride-components containing BGs on dentine. The abrasive powder was prepared by the melt-quench route. The cutting capacity of the developed BG particles on dentine through air abrasion was assessed by white light profilometry and scanning electron microscope. The results showed that these novel bioactive glasses were highly degradable and could initiate fast fluorapatite deposition in a biological environment. Moreover, in the case of high fluoride-containing BGs, CaF_2_ formation was also observed. BGs with a higher chloride content degraded faster. The hardness of these BGs could be adjusted to the requirements by changing the chloride concentration in the structure, thus they could discerningly cut various dental tissues. For clinical application, it is of great importance for caries preparation, while a softer glass is more suitable for tooth cleaning. Figure 5 demonstrates the most common application possibilities, both in hard tissue engineering as well as in dentistry.

### 5.4. Bioactive Glasses as Coating Materials (for Orthopedic and Dental Implants)

It is reported that the bone tissue regeneration process works through so-called biophysical stimulation processes [213,214]. To boost this stimulation effect and to accelerate integration of the load-bearing implants (either metallic or ceramic), the best method is depositing highly bioactive layers onto their surfaces [122,215].

So far, many different surface modifications and coatings with various thicknesses have been developed and tested for faster bone regeneration and to enhance the connection between the implants and the adjacent soft and hard tissues [47,122,216,217]. 

It is discussed that the surface morphology, such as roughness, is as much an important factor in the implants’ integration as the chemical composition and structure of the coating itself, through the support of bone cell attachment [216]. The most commonly used deposition techniques so far are airbrushing [217], spin-coating [218], dip-coating [219], and electrophoretic deposition [220,221], which are using sol-gel precursors and subsequent heat treatment; as well as enamelling [222], plasma-spraying [223]; physical vapour depositions (PVD), such as pulsed laser deposition (PLD), pulsed electron deposition (PED); and radio-frequency magnetron sputtering (RF-MS) [224], micro-arc oxidation [225,226], and thermal spraying [227] which use powder precursors directly applied to the substrates’ surfaces. In order for the coating to be efficient, several factors have to be considered, such as the thermal expansion coefficient values of bioactive glass and the substrate which must be quite similar to avoid cracks and peeling off during the heat-treatment. However, applying an intermediate bonding layer could be a solution to this problem [228]. The thickness of the layer is also important, because the thicker the layer, is more likely delamination is to occur. Lastly, the sintering parameters themselves can profoundly affect the mechanical properties as well as the strength of attachment of the coatings [229].

The biological performance of the different bioactive glass coatings was assessed in many reports, in vitro and in vivo alike [230,231,232,233,234,235,236,237]. After soaking in body fluids, apatite layer formation was observed [230,231].

In vivo tests are also important to check their cytotoxic nature or their cytocompatibility. In these cases, the bioactive glass coated implants were tested in different animals for a long period [232,233,234,235,236,237]. They similarly ascertained that the feature of the bone-bioglass surface connection and reactions is crucially important for the complete integration of implant, as well as for its long-term applicability.

Aside from the many review papers [238,239,240,241] that give a useful and exhaustive insight into the achievements so far in present area, there is intensive ongoing research on the further development of both the composition of the bioactive glass composite coatings and optimization of their preparation method, which is the key factor of the goodness of the adherence. One of the perfect examples is the up-to-date work of Canas et al. [242], wherein an innovative and appropriately optimized bioactive glass material was applied as a coating on stainless steel substrates by plasma spraying technique. The powder for the coatings was prepared by melt-quenching. The obtained coatings were compact with sufficient porosity and bioactivity, according to the in vitro tests. Figure 6 demonstrates the already existing application possibilities to date.

## 6. Conclusions Remarks and Future Outlook

As a summary of the present review, it can be clearly recognized that there is an immense need for high quality, bioactive glasses that meet the high standard for clinical and even industrial applications. To answer to this demand, there is intensive research and competition for the of development bioglasses with better and better properties. According to the application field and the intended usage, the bioglasses can be made and tailored in various forms.

However, there are many issues that have not been solved yet. For example, the standardization of the procedures and parameters which are essential for reproducibility. The appropriate surface roughness is also crucial for the proper osseointegration. Hence, it also needs to be adjusted to the given requirements for both coatings and scaffolds. For coatings, the most critical issue is the thickness. The thickness determines the degradation time and rate as well as the amount of the doping elements/drugs, if applied. However, according to many experiments and even clinical trials, the thick layers are more prone to peel off from the surface of substrate, causing implants’ failure and serious negative side effect. Therefore, in this case, a thinner layer is advisable. On the other hand, for the scaffold materials, the forming of an appropriate and even pore size with highly interconnected pores (that most resembles the structure of human bones) is the most difficult to achieve. There are many attempts to make scaffolds with perfect pore sizes, distributions, and interconnectivity by using 3D printing technique or by applying templates during the formation of mesoporous scaffolds. When the scaffolds are intended to be used as bone substitutes or grafts, they might need to be formed into various shapes adaptable to the complex nature of bone defects and, in addition, their mechanical properties must match that of the bones. These complex shapes can be the most efficiently obtained by 3D printing. Unfortunately, the bioactive glasses developed so far still do not have sufficient mechanical strength and are fragile. These facts hinder their applicability in load-bearing bone defects, so these problems also need to be addressed. There are attempts to strengthen the mechanical properties of porous bioglass structures by incorporating different metal oxides, such as magnesium-, zinc-, zirconium, or aluminum oxides [243].

For safer applicability, the exact steps of the interactions and reactions between the different seeded cell lines and bioactive glass materials in the various and dynamic biological environment within the body need to be explored and better understood, since the cells might react differently in dinamic conditions, compared to the static conditions of in vitro tests. Moreover, the different types of implants (such as scaffolds, grafts, coatings, or even bulk implant materials) are supposed to be used in different body parts providing completely different environment and conditions. 

Owing to their unique and tailorable micro- and nanostructure, as well as arbitrarily variable composition, the bioactive glasses can be easily doped with elements and drugs which are released controllably and locally at the aimed site, thus providing other advantages to these materials for bone- tissue regeneration or even fighting against infections.

The application of bioactive glasses in soft tissue engineering is a relatively new emerging area with great possibilities. These materials have huge potential in healing soft tissue damages. The soft tissues can be the skin (accelerating the wound healing), or even muscular, nervous, epithelial, cardiac, lung, and cartilaginous tissues. In this specific research field, the key factor is the mechanical and structural properties of the BG composites to be used, since they have to properly imitate the structure of targeted soft tissues in order for a complete healing without complications. According to the current state of research, there is still a dissimilarity between the mechanical strength of the developed BG composites so far and the targeted soft tissues. Hence, many experiments for achieving the optimum composition, structure, strength, and biodegradability, as well as more in vitro and in vivo tests, are still required to develop further to clinical trials or even commercialization. The proper structure of these materials can be most easily obtained by modern and continuously developing additive manufacturing technology, such as 3D or even 4D printing. These unique and flexible techniques allow for customization of the materials with complex shape and structure which can be easily altered according to any specific requirements. In addition, they create less waste or by-products. The different BGs in this case can be produced in the form of fibers and scaffolds when mixed with the chosen biopolymers. In some cases, the injectability of these composites is also a requirement. However, to date, obtaining the perfect BG-composite structures is still a great challenge due to the lack of sufficient number of in vivo tests and absence of any clinical trials so far. The specific biological reactions (dissolution, ions release and their effect on the body) of BGs in contact with the damaged soft tissues in the dynamic biological environment are still yet to be thoroughly investigated. In other words, despite their immense number of benefits, these novel and modified new generation bioglasses are still not introduced to the market because of the above mentioned lack of clinical trials, data, and reported post-operative follow ups. In addition, during the development of new generation bioglass with a unique composition and properties another key factor is its biodegradability rate, which must be tailored to the actual usage (such as site of the body and the specific application fields), since a faster degradation for a dental coating or implant than that for a bone graft scaffold may be required. For example, some scaffolds’ degradation has to be exceptionally slow to allow for tissue or cartilage damage regeneration. Performing any clinical trials also faces numerous obstacles, such as a sufficient number of patients and authorizations of health organizations. It requires tremendous effort, work, and also administration. In vivo animal investigation would be a reasonable intermediate step before human experiments, and their results can be more reliably used in the next step.

## Figures and Tables

**Figure 1 nanomaterials-13-02287-f001:**
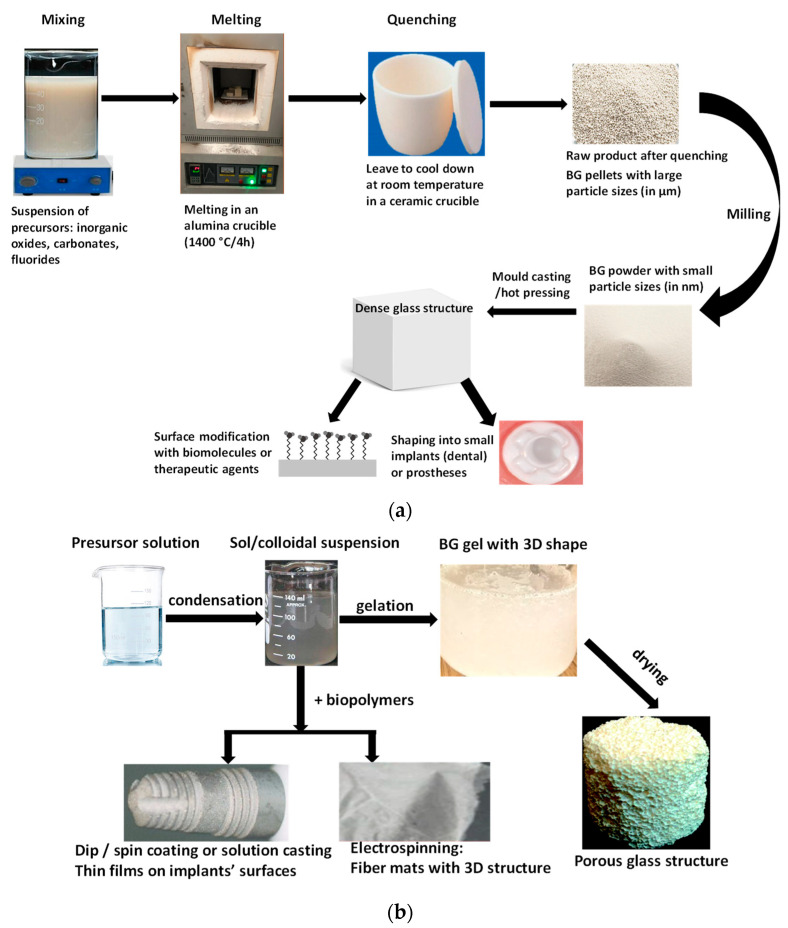
Schematic detailed illustration of the two main preparation processes, such as melt-Quench (**a**) and sol-gel route (**b**).

**Figure 2 nanomaterials-13-02287-f002:**
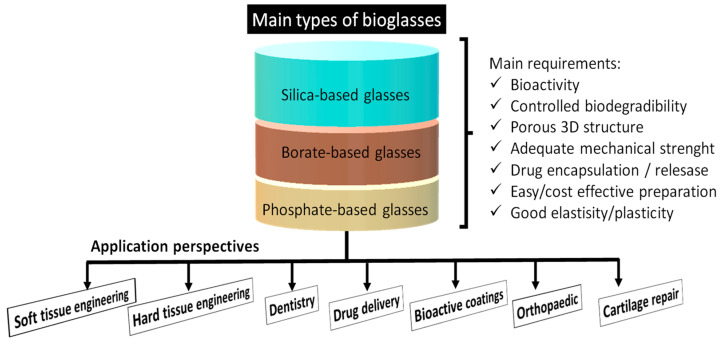
The three commonly developed and researched bioglass types, their required properties, as well as their applications in different biomedical fields.

**Figure 3 nanomaterials-13-02287-f003:**
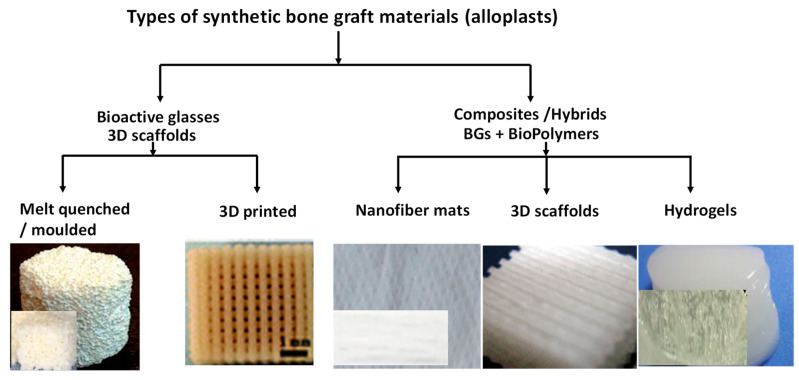
Typical forms of synthetic bone grafts (alloplasts).

**Figure 4 nanomaterials-13-02287-f004:**
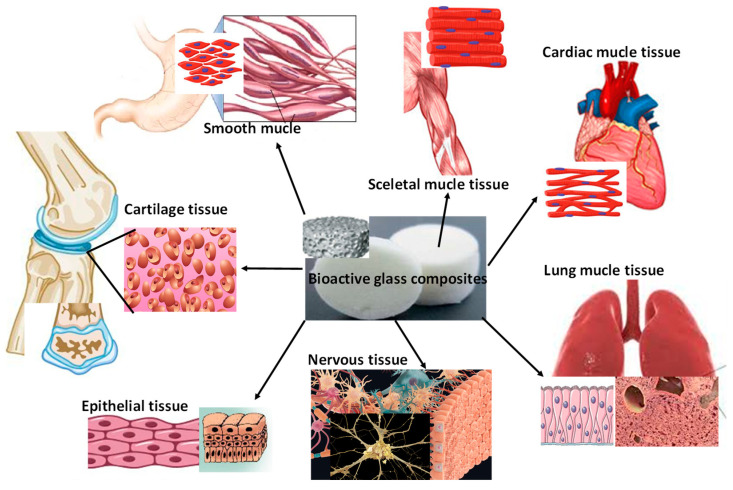
Schematic illustration of the potential usage of novel bioactive glasses in soft tissue engineering.

**Figure 5 nanomaterials-13-02287-f005:**
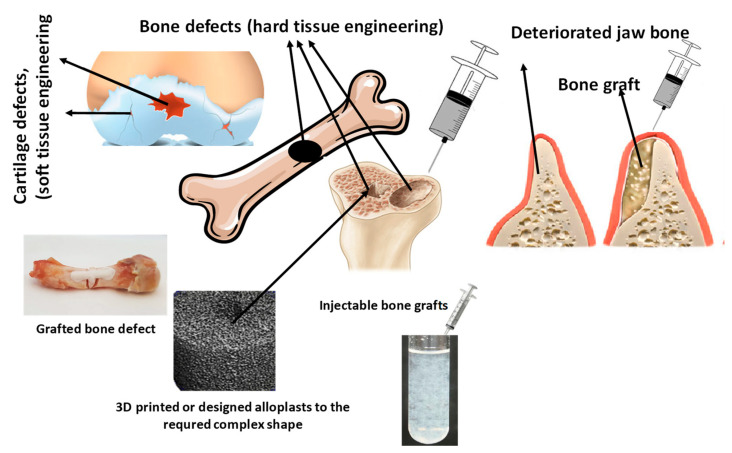
Schematic illustration of the practical usage of different bone grafts.

**Figure 6 nanomaterials-13-02287-f006:**
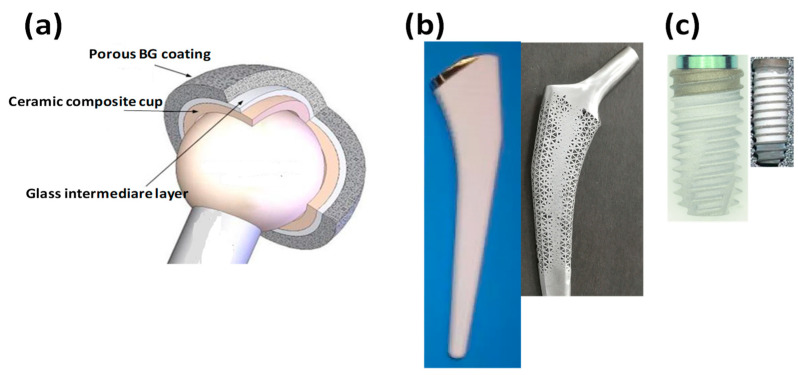
Illustration of the application possibilities of bioactive glass coatings on load-bearing implants (**a**) schematic draw on the construction, (**b**) coatings on hip implants, (**c**) coatings on dental implant fixtures.

**Table 1 nanomaterials-13-02287-t001:** Summarization of the main advantages and disadvantages of the different BGs according to the performed thorough literature survey.

Types of BGs:	Silica-Based BGs	Borate-Based BGs	Phosphate-Based BGs
Benefits	Easy production with both melt-quench and sol-gel methods Can be shaped in various forms Composition can be arbitrarily changed Doping with bioactive elements is easy to achieve Strong BG-Bone bond Highly bioactive	Composition can be arbitrarily changed Can be shaped in various forms Due to the lower network connectivity, they can completely transform into cHAp. Boron substitution enhance the mechanical properties of the BG scaffold Decreased crystallization rate High bioactivity Strong BG-Bone bond Doping with bioactive elements is easy to achieve	Can be shaped in various forms Higher and more easily controllable degradation rate Wider solubility range and can entirely dissolve in aqueous environments Stimulate bone formation Highly bioactive Highly porous materials can be prepared with large specific surface area Decreases crystallization rate Complete and fast transformation into cHAp in biological conditions Strong BG-Bone bond Doping with bioactive elements is easy to achieve
Drawbacks	Mechanical and chemical performance of these materials is highly dependent on the aging mechanisms Undesirable crystallization can occur during improper sintering (poor sinterability) Relatively slow degradation The conversion of BGs to cHAp is not complete, the cHAp layer formation on the surface is slow	The fast degradation in biological environment limits their applicability The production by the sol-gel method is challenging, because of the difficult chemistry and reaction mechanisms Poor mechanical properties Cannot be achieved fully 3D structures (owing to the coordination number of boron)–weak network interconnection	The fast degradation in biological environment limits their applicability The production by the sol-gel method is challenging, because of the difficult chemistry and reaction mechanisms Poor mechanical properties

**Table 2 nanomaterials-13-02287-t002:** Summarization of the most common doping agents (bioactive elements, therapeutic agents, drugs) and their biological effect. The Ca, P, Si, and B components are excluded, since they are the constituents of the bulk BGs.

Dopants		Effects	Main Role
Bioactive trace elements	Magnesium	✓ Essential trace element in nerves, muscles, necessary for the immune system. ✓ Increase the mechanical strength of bones ✓ Increase the bioactivity of bioglasses/ceramics	Osteogenesis/Osteoinductivity
	Strontium	✓ Enhance the metabolic activity in osteoblasts ✓ Boost osteoclast activity ✓ Increase the bioactivity of bioglasses/ceramics	
	Fluoride	✓ Increase the mechanical strength of bones/teeth (used mainly in dentistry)	
	Manganese	✓ Essential trace element in all organs, ✓ used by many enzymes, ✓ takes role in several biological processes and bone formation	
	Iron	✓ Trace element, crucial for blood production, blood vessel formation. ✓ Has a huge role in many metabolic processes	
	Zinc	✓ Essential trace element, necessary of the normal function of countless enzymes and can be found in high concentration in bones	
	Cobalt	✓ Component of B12 vitamin, ✓ supports the nervous system, ✓ Essential for cell metabolisms	Angiogenesis
Therapeutic elements/agents	Zinc oxide	✓ Has biocompatible, antibacterial, antifungal, antiviral, and even anticancer effects	Antibacterial anti-inflammatory
	Copper	✓ Protect the cardiovascular system, increase lung elasticity, promote neovascularization, neuroendocrine function, and iron metabolism. ✓ Catalytic reagent of several enzymes and proteins. ✓ Facilitate bone fracture healing ✓ Activate bone metabolisms	Angiogenesis osteogenesis
	Copper oxide	✓ Attach to the bacteria cell membrane, causing a drastic change in it, (membrane integrity) which results in bacteria death.	Antibacterial Anti-fungicide
	Cerium	✓ Potential pharmacological agents for gene activation. ✓ Inhibiting the functions of osteoclast cells	Osteogenesis
	Cerium oxide	✓ Increase the levels of ROS in bacteria, ✓ Damage their proteins causing death of the bacteria	Antibacterial
	Gallium	✓ Inhibits the bone cell-resorbing ability of osteoclasts as well as used as therapeutic agent in cancer and dysfunctions of calcium and bone metabolism. ✓ Fight against infectious microorganisms. ✓ Disturbs the iron-dependent proliferations of cancer cells.	Osteogenesis. Antibacterial
	Silver	✓ Wide spectrum antibiotic element, did not induce bacteria resistance. ✓ React with bacterial or fungal cell membranes. ✓ Causes malfunctioned mitochondrial mechanisms and therefore leakage through the bacteria cell membranes.	Antibacterial, Anti-fungicide. Anticancer, Antioxidant, Anti-inflammatory, Wound healing
	Lithium	✓ Low content of Li is advantageous for the living organs, since it activates Wnt-catenin signal route, ✓ promote the act of some genes and proteins as well as affect COL1 and ALP action mechanisms. ✓ On the other hand, high concentration of it can cause cytotoxicity and has negative side effects	Osteogenesis
Drugs/Antibiotics	Amoxicillin	✓ Semi-synthetic antibiotic	antibacterial
	Ricampicin	✓ Semi-synthetic antibiotic	
	Ciprofloxacin	✓ Second generation antibiotic	
	Levofloxacin	✓ Synthetic antibiotic	
	Gentamicin	✓ Antibiotic aminoglycoside	
	Vancomycin	✓ Antibiotic –glycopeptide, effective against Gram+ bacteria	
	Tobramycin	✓ Aminoglycoside antibiotic derived from Streptomyces tenebrarius that is used to treat various types of bacterial infections	
	Ceftriaxone	✓ Third-generation antibiotics	
	Sulbactam sodium (CFS)	✓ Third-generation antibiotics	
	Flufenamic acid	✓ Nonsteroidal anti-inflammatory drugs	Anti-inflammatory
	Ibuprofen	✓ nonsteroidal anti-inflammatory drug (NSAID) to treat mild to moderate pain	Pain killer
	Alendronic acid	✓ Bisphosphonate, prevent and fight against osteoporosis	Anti-osteoporosis

## Data Availability

Not applicable.

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
