# Peer review of "Advanced Bioactive Glasses: The Newest Achievements and Breakthroughs in the Area"

_nanomaterials, 2023, doi:10.3390/nano13162287_

Round 1
Reviewer 1 Report
The topic covered by this review is important and relevant, and the field of bioactive glasses as hard and soft tissue regeneration has attracted more research attention. The authors did a lot of work to review relevant studies, however, the current manuscript lacked focus and scientific depth, especially in the section 6. Conclusion remarks and future outlook, just repeated the point views mentioned before. Necessary revisions are needed before considering for publishing.
1、The title of 5.1. Bioglasses as artificial bone replacements (soft and hard tissue engineering) is confusing. In this section, bioactive glasses used for bone tissue regeneration were reviewed, however, no soft tissue regeneration mentioned.
2、In this manuscript, bioactive glasses and bioglasses were both used, which is also confusing. As I know, bioglass is refer to 45S5.
3、A lot of researches have been done in the field of bioactive glasses used in soft tissue regeneration beyond wound healing, this part is missing in this review.
4、In the section Conclusion remarks and future outlook, just repeated the point views mentioned before. As a review, more scientific depth and outlook are needed.
5、To improve readability, more figures cited from published papers are needed in this review.
The expression should be improved.
Author Response
Dear Reviewer 1,
1、The title of 5.1. Bioglasses as artificial bone replacements (soft and hard tissue engineering) is confusing. In this section, bioactive glasses used for bone tissue regeneration were reviewed, however, no soft tissue regeneration mentioned.
Answer:
Thank you for the constructive comment, you are totally right. We have corrected the title to “5.1. Bioactive glasses in soft and hard tissue engineering” to be more clear and also made the necessary corrections in the text. Moreover, we have looked through thoroughly the available literature again for the current state of bioactive glass usage in soft tissue engineering and supplemented the text with the most relevant papers, data. Citations are also added.
2、In this manuscript, bioactive glasses and bioglasses were both used, which is also confusing. As I know, bioglass is refer to 45S5.
Answer:
Thank you for the observation. We have corrected the bioglass phrase to bioactive glass where it was necessary. Yes, originally the bioglass refer to the commercial 45S5, which was first developed by Larry Hench and his group at the University of Florida in 1969 [Hench, L.L.; Splinter, R.J.; Allen, W.; Greenlee, T. Bonding mechanisms at the interface of ceramic prosthetic materials. J. Biomed. Mater. Res. Part A 1971, 5, 117–141.]. Since then there was a huge development in the BGs, and now exist second and even third generation BGs with very diverse constitution and properties.
We can say that many types of BG-based products currently available in the market, such as the original 45S5 Bioglass®, which are mainly used for hard tissue engineering to heal bone damages in orthopedics and dentistry. The so-called bioactive glasses (BG) are regarded as a third-generation biomaterials, and have great attention and potential in various fields.
3、A lot of researches have been done in the field of bioactive glasses used in soft tissue regeneration beyond wound healing, this part is missing in this review.
Answer:
We have added the missing research data and achievements so far in the text, according to the suggestion.
4、In the section Conclusion remarks and future outlook, just repeated the point views mentioned before. As a review, more scientific depth and outlook are needed.
Answer:
We have made some supplementation to the Section 6 after browsing through the existing literature data on this specific topic. We made the addition to the best of our knowledge, and based on the already available literature data either in research papers or reviews.
5、To improve readability, more figures cited from published papers are needed in this review.
Answer:
We have added one more Figure to the manuscript, regarding the soft tissue engineering. In our paper, we have used our work of art using free internet source avoiding copyright issues. We sincerely hope, you find this sufficient and deem our paper acceptable for publication.
Reviewer 2 Report
The review can be published after minor revisions.

Author Response
Dear Reviewer 2,
1.
Page 3 , 83-86 The phrase: “On the other hand, the sodium oxide content in the bioglass causes a higher inclination towards crystallization, limiting their shaping into different forms, and has some cytotoxic effect as well [21] which is caused by the dissolution of alkali ions into the physiological solutions or body fluids.” must be extended. The authors must explain what crystalline phases appears in the glass structure and how are they toxic for the human body.
Answer:
Thank you for the constructive comment, we have added the requested information into the text of the manuscript.
2.
Can the authors make an assessment of crystalline phase crystallization or cytotoxicity in case of ZnO, ZrO2, CeO2 or MnO addition? The mentioned oxides can help crystallization as can be seen in References :
Influence of Ceria Addition on Crystallization Behavior and Properties of Mesoporous Bioactive Glasses in the SiO2–CaO–P2O5–CeO2 System, EM Anghel, S Petrescu, OC Mocioiu, JP Cusu, I Atkinson, Gels 8 (6), 344, 2022.
Cerium-containing mesoporous bioactive glasses (MBGs)-derived scaffolds with drug delivery capability for potential tissue engineering applications, I Atkinson, AM Seciu-Grama, S Petrescu, D Culita, OC Mocioiu, M Voicescu, R -A Mitran, D Lincu, A -M Prelipcean, O Craciunescu, Pharmaceutics 14 (6), 1169, 2022
Remineralization of natural tooth enamel in artificial saliva environment, A Zaharia, VG Plescan, I Atkinson, OC Mocioiu, A Cantaragiu, V Musat, Rev Chim 68 (3), 510-514, 2017
Influence of ZnO addition on the structural, in vitro behavior and antimicrobial activity of sol–gel derived CaO–P2O5–SiO2 bioactive glasses, I Atkinson, EM Anghel, L Predoana, OC Mocioiu, L Jecu, I Raut, C Munteanu, D Culita, M Zaharescu, Ceramics International 42 (2), 3033-3045, 2016
Answer:
We have made corrections in the text, according to the suggestions, however, according to the assessment of crystalline phase and crystallization or cytotoxicity in case of ZnO, ZrO2, CeO2 or MnO we could only use the given literature data. We have added the relevant suggested citations and the results described in them to the manuscript.
3.
The authors must check and correct the way they write oxides. They must use subscript such as: 47.5SiO2-10Na2O-10K2O-10MgO-20CaO- 384 2.5P2O5
Answer:
Thank you for the observation, we corrected the forms of compositions